# Cold Atmospheric Plasma Promotes the Immunoreactivity of Granulocytes In Vitro

**DOI:** 10.3390/biom11060902

**Published:** 2021-06-17

**Authors:** Laura S. Kupke, Stephanie Arndt, Simon Lenzer, Sophia Metz, Petra Unger, Julia L. Zimmermann, Anja-Katrin Bosserhoff, Michael Gruber, Sigrid Karrer

**Affiliations:** 1Department of Anesthesiology, University Hospital Regensburg, 93053 Regensburg, Germany; Laura.Weidner@stud.uni-regensburg.de (L.S.K.); Simon1.Lenzer@stud.uni-regensburg.de (S.L.); Sophia.Metz@ukr.de (S.M.); michael.gruber@ukr.de (M.G.); 2Department of Dermatology, University Hospital Regensburg, 93053 Regensburg, Germany; petra.unger@ukr.de (P.U.); sigrid.karrer@ukr.de (S.K.); 3Terraplasma Medical GmbH, 85748 Garching, Germany; zimmermann@terraplasma.com; 4Emil-Fischer-Center, Institute of Biochemistry, University of Erlangen-Nuernberg (FAU), 91054 Erlangen, Germany; anja.bosserhoff@fau.de

**Keywords:** cold atmospheric plasma, wound treatment, granulocytes, live-cell imaging, NETosis, flow cytometry

## Abstract

Cold atmospheric plasma (CAP) reduces bacteria and interacts with tissues and cells, thus improving wound healing. The CAP-related induction of neutrophils was recently described in stained sections of wound tissue in mice. Consequently, this study aimed to examine the functionality of human polymorphonuclear cells (PMN)/granulocytes through either a plasma-treated solution (PTS) or the direct CAP treatment with different plasma modes and treatment durations. PTS analysis yielded mode-dependent differences in the production of reactive oxygen species (ROS) and reactive nitrogen species (RNS) after CAP treatment. Live-cell imaging did not show any chemo-attractive or NETosis-inducing effect on PMNs treated with PTS. The time to maximum ROS production (T_max_ROS) in PMNs was reduced by PTS and direct CAP treatment. PMNs directly treated with CAP showed an altered cell migration dependent on the treatment duration as well as decreased T_max_ROS without inducing apoptosis. Additionally, flow cytometry showed enhanced integrin and selectin expression, as a marker of activation, on PMN surfaces. In conclusion, the modification of PMN immunoreactivity may be a main supporting mechanism for CAP-induced improvement in wound healing.

## 1. Introduction

Cold atmospheric plasma (CAP) is a partially ionized gas. It contains electrons, ions, reactive oxygen species (ROS), reactive nitrogen species (RNS), and small amounts of optical emission in ultra-violet and infrared radiation ranges. CAP is becoming increasingly important in medical applications because of its noninvasiveness and fast administration as well as its broad range of applications [1]. The effects of CAP are either transmitted via direct treatment or mediated by plasma-treated solutions (PTS) [2,3,4,5]. Due to its antibacterial effect, direct CAP treatment is used for contactless disinfection of surfaces and wounds, for example [6,7]. CAP is also known for its positive interactions with tissue, promoting VEGF and TGF-β protein expression [8], and its stimulating effects on eukaryotic cells. CAP induces the expression of β-defensins and inflammatory cytokines in keratinocytes and promotes proliferation and migration [9,10,11,12]. In addition, CAP stimulates the fibroblast production of cytokines and growth factors as well as their migration [13,14,15]. Recently, a CAP-related increase in the number of granulocytes was observed in stained sections of wound tissue in mice [13].

Human polymorphonuclear cells (PMN)/granulocytes are circulating cells of the innate immune system. Among others, PMNs play a central role in the first inflammatory response after tissue injury and infection, although their persistence is associated with chronic lesions [16,17]. In the case of inflammation, PMNs are initially activated by bacterial mediators or the endogenous pyrogen tumor necrosis factor-alpha (TNFα) [18]. Thereupon, PMNs migrate to the site of injury by recognizing damage-associated molecular patterns transmitted from damaged cells or by chemo-attractive signals such as interleukin-8 [16,19]. After infiltration, PMNs act as phagocytes, release antimicrobial peptides, ROS, and cytotoxic enzymes during a process termed respiratory burst. After that, they produce neutrophil extracellular traps (NETs) during a process called NETosis to eradicate microbes and resolve inflammation [16,17]. In addition, PMNs promote angiogenesis, fibroblast proliferation, and monocyte recruitment in the sense of transitioning from the inflammatory to the proliferative phase of wound healing [16]. The fact that PMNs are essential for appropriate tissue repair becomes apparent in neutropenic patients, who often have wound healing deficits [20]. Furthermore, PMN functions may be modified with several drugs, leading to clinically relevant immunomodulation [21,22,23]. In these studies, the typical time course of PMN reactions is described as follows. Initially, the granulocytes are activated. This is followed by migration with a simultaneous increase in ROS production. After maximum ROS production, migration ends and is replaced by NETosis. In case of increased activation, for example by certain drugs, the subsequent processes are expected to be faster, but not stronger. Although the processes are chronological, they do not necessarily have to be directly related to each other.

As cells important for wound closure, fibroblasts and keratinocytes benefit from CAP treatment. Yet, only little is known about the CAP impact on PMNs, which are the first responders to infection and tissue damage and are responsible for effective wound healing. Additionally, CAP treatment of wounds seems to be the most important interface of PMNs and CAP. Therefore, this study aimed to investigate the CAP-induced effects on PMN functions in vitro, such as chemotactic migration, ROS production, NETosis, and expression of surface antigens.

## 2. Materials and Methods

### 2.1. Granulocyte Preparation

Whole lithium heparin-anticoagulated blood samples were taken from healthy donors after informed consent. In this study, we used blood samples from 25 donors distributed among different experiments. PMNs were isolated by density gradient centrifugation (756× *g*) at ambient temperature for 20 min using PBMC Spin medium stratified on a Leuko Spin medium (pluriSelect Life Science, Leipzig, Germany). Dulbecco’s phosphate-buffered saline (DPBS, Sigma Aldrich GmbH, Steinheim, Germany) served for washing the PMNs. After that, the cells were resuspended in RPMI 1640 (Pan-Biotech GmbH, Aidenbach, Germany) dosed with 10% fetal bovine serum (FBS, Sigma-Aldrich GmbH, Steinheim, Germany) at a concentration of 18 × 10^6^ cells/mL. All experiments were approved by the Ethics Committee of the University of Regensburg, Germany (Vote no: 19-1569-101).

### 2.2. Plasma Device and Granulocyte Treatment

A prototype of the plasma care^®^ device (Terraplasma GmbH, Garching, Germany), a surface micro-discharge (SMD) plasma source, was used for CAP treatment (Appendix A). This prototype enables changes in frequency between oxygen (4 kHz) and a nitrogen (8 kHz) mode. Usage of the high voltage of 3.5 kV provokes millimeter-sized micro-discharges into the plasma source unit. The unit consists of a high-voltage electrode and a dielectric and a grounded structured electrode (Appendix A), which subsequently produce plasma components alterable by frequency and voltage. To guarantee a standardized distance between the device and the Petri dish (35 mm, Corning, Merck, Darmstadt, Germany), the device was placed onto a spacer (Appendix A). Simultaneously, this ensures an isolated treatment area. Ozone (O_3_) values in parts per million (ppm) produced by the plasma care^®^ device using variable frequencies of 4 kHz and 8 kHz, 3.5 kV, and a 5 min treatment time are shown in Appendix A. O_3_ generated by the SMD device was measured in a confined volume by ultraviolet (UV) absorption spectroscopy according to [24] and provided to us by Terraplasma GmbH. O_3_ is quenched by nitric oxide and nitrogen dioxide [24], as observed in the 8 kHz treatment mode.

The PMN samples were used either untreated or split into aliquots for CAP treatment, altering the plasma mode and treatment duration depending on the experimental setup. For the creation of PTS, 2.5 mL of RPMI 1640 dosed with 10% FBS was put into a Petri dish with a magnetic stir bar for stirring at 300 rounds per minute. For direct CAP treatment, 2.5 mL of the isolated PMNs suspended in DPBS were placed into a Petri dish before resuspension in RPMI 1640. CAP treatment parameters were a frequency of 4 kHz or 8 kHz and a duration of 2 min or 5 min to investigate the mode- and dose-dependent effects of CAP. A corresponding control group remained untreated.

### 2.3. ROS and RNS Measurement in PTS

For ROS detection, 10 µM dihydrorhodamine 123 (DHR123, Sigma Aldrich GmbH, Steinheim, Germany) was solubilized in 100 µL of PTS immediately after CAP treatment and transferred back to a black 96-well plate (Greiner Bio-One GmbH, Frickenhausen, Germany). Fluorescence was measured at Ex = 505 nm and Em = 534 nm. H_2_O_2_ was quantified with a Fluorimetric Hydrogen Peroxide Assay Kit (Sigma Aldrich GmbH, Steinheim, Germany), and fluorescence was measured at Ex = 540 nm and Em = 590 nm. NO_2_^−^ and NO_3_^−^ concentrations were determined by using the colorimetric Nitrite/Nitrate Assay Kit (Sigma Aldrich GmbH, Steinheim, Germany) to detect nitric oxide metabolites at 540 nm absorbance. Fluorescence was measured with a plate reader (Varioscan Flash, Thermo Fisher, Schwerte, Germany), and the Assay Kits were used as specified by the manufacturer.

### 2.4. Evaluation of Cell Migration, ROS Production, and NETosis by Live Cell Microscopy

The experimental setup was based on previous experiments published in [22,25,26]. To observe PMN behavior, a chemotactic assay was performed using 3D-µ-slides (ibidi© GmbH, Martinsried, Germany) according to the manufacturer’s instructions. Every slide consisted of three separated channels each, of which were bounded by two reservoirs (Appendix A). This design enabled the analysis of two different CAP treatment methods next to an untreated control group. The prepared PMNs were suspended in type I collagen gel (1.5 mg/mL PureCol, Advanced BioMatrix, Carlsbad, CA, USA) supplemented with fluorescent stains and filled into the channels. Intracellular ROS production was visualized with 1 µM DHR123. For the detection of NET formation, 0.5 µg/mL 4′,6-diamidin-2-phenylindol (DAPI, Sigma-Aldrich GmbH, Steinheim, Germany) was used for staining extracellular DNA. After incubation under humid conditions at 37 °C and 5% CO_2_ for 30 min, RPMI 1640 dosed with 10% FBS was poured into each reservoir. To create chemotactic PMN movement, the chemoattractant N-formyl-methionine-leucyl-phenylalanine (fMLP, 10 nM, Sigma Aldrich GmbH, Steinheim, Germany) was added to the reservoirs on the left side of the channels. Depending on the experimental setup, the PMNs or RPMI 1640 dosed with 10% FBS were CAP treated previously, or the chemo-attractant fMLP was replaced with PTS (Appendix A).

After preparation, the slide was viewed by live-cell imaging using a Leica DMi8 microscope (Leica Microsystems, Wetzlar, Germany). The microscope is equipped with a motorized adjustable stage, a Leica DFC9000 camera (Leica Microsystems, Wetzlar, Germany), a pE-4000 light source (CoolLED, SP10 5NY, Andover, England), and a stage top incubator (ibidi© GmbH, Martinsried, Germany) to ensure stable conditions at 37 °C and 5% CO_2_. Phase-contrast and fluorescence images were recorded automatically every 30 s for 10 h with Leica Application Suite X software (version 3.7.3.23245, Leica Microsystems, Wetzlar, Germany). Two example videos showing ROS-production/NETosis and migration are provided as Appendix A.

### 2.5. Analysis of the Microscopic Image Sequences from Cell Migration, ROS Production, and NETosis

Microscopic image sequences from Section 2.4 were subsequently analyzed with Imaris software (version 9.0.2, bitplane, Zurich, Switzerland).

To quantify cell migration, the software used preassigned 30-min time periods of phase-contrast images for the semi-automated tracking of migrating cells. The calculated track data for every moving cell encompassed the Track Displacement (TD, TDX, TDY, and Euclidean track, in total and divided into x- and y-directed migration) and the Track Length (TL, accumulated migration). Cells with tracks less than 25 µm and under a 900 s duration were excluded from the analysis to eliminate non-migrating cells and particles.

Fluorescence images provided the basis for ROS production and NETosis analyses by processing the total fluorescent areas per time point. After exportation into the Excel software program (Microsoft Excel version 16.45), a third-degree polynomial trendline was adjusted to the characteristic parabolic course plotted by time vs. summed up surface area to calculate T_max_ROS with the zero point of the first derivate. NETosis data resulted in sigmoidal curves. The calculation of ET_50_NETosis was therefore processed with Phoenix 64 software (version 8.0.0, Certara Inc., New York, NY, USA).

### 2.6. Measurement of Cell-Surface Antigens, Respiratory Burst, and Cell-Membrane Permeability by Flow Cytometry

Flow cytometry (FACSCalibur, BD Biosciences, San Jose, CA, USA) was applied along with the CellQuest Pro software (version 5.2, BD Biosciences, San Jose, CA, USA) to assess cell-surface antigen expression, respiratory burst, and cell-membrane permeability. PMNs were treated identically to the samples used in live-cell imaging and used in two technical replicates for measurements 2 h and 6 h after CAP treatment to observe changes over time.

Cell-surface antigen expression was detected by means of the fluorochrome-conjugated antibodies CD11b (ICRF44, PE-conjugated, BioLegend, San Diego, CA, USA), CD62L (DREG-56, FITC conjugated, BioLegend), and CD66b (G10F5, APC conjugated, BioLegend).

To observe ROS production by respiratory burst, cells were incubated in DPBS, DHR123 (10 µM), and seminaphtharhodafluor (SNARF, 10 µM, Invitrogen, Eugene, OR, USA). SNARF was used as an additional internal control with no further relevance to this study [27]. The oxidative burst was caused either by fMLP (10 µM) and tumor necrosis factor-alpha (TNFα, 1 µg/mL, Pepro Tech Inc., Rocky Hill, CT, USA) or by phorbol-12-myristate-13-acetate (PMA, 10 µM, Sigma Aldrich GmbH, Steinheim, Germany), which served as a positive control.

Propidium iodide (PI, 1.5 mM, Invitrogen, Eugene, USA) was added to assess cell-membrane permeability.

The data analysis was implemented using the FlowJo software (version 10.7.1, FlowJo LLC, Ashland, OR, USA).

### 2.7. Statistical Analysis

After collecting data from live-cell imaging and flow cytometry in Excel, GraphPad Prism (version 9, GraphPad Software Inc., San Diego, CA, USA) was used for further statistical calculations. Contingent on the normal distribution tested with the Kolmogorov-Smirnov-test and paired or non-matching data, analysis of variance (ANOVA) was conducted using the Friedman test or the Kruskal–Wallis test. Multiple comparison testing was carried out afterward. Results are expressed as a median with an interquartile range (IQR). Detailed information about the results can be found in Appendix A. A *p*-value < 0.05 was considered statistically significant (labeling in figures: * *p* < 0.05, ** *p* < 0.01, *** *p* < 0.001, **** *p* < 0.0001, ns: not significant).

## 3. Results

This study aimed to examine the modified activity and functionality of PMNs through either PTS or direct CAP treatment with frequencies of either 4 or 8 kHz and treatment durations of 2 or 5 min to reveal a potential supporting mechanism for improving wound healing by CAP.

### 3.1. Modified Migration of CAP-Treated PMNs without Any Chemo-Attractive PTS Effect

Granulocyte migration was analyzed by tracking the migrating cells with live-cell imaging microscopy. Firstly, a potential chemo-attractive PTS effect was investigated. Secondly, the PMNs were directly CAP-treated before microscopy, and the migration that may have been altered as a result was observed to reveal CAP impact on PMN migration.

The chemo-attractive effect of PTS was tested by replacing the chemo-attractant fMLP in the left reservoirs of the 3D-µ-slide with 2 min or 5 min of 4 kHz PTS. The first 60 min of observation in the control group showed a chemotactic movement towards the chemo-attractant. However, Track Length (Figure 1A(I)) and Track Displacement X (Figure 1A(II)) were significantly decreased when fMLP was exchanged with 2 min or 5 min of PTS. These findings led to the assumption that PTS has no chemo-attractive effect on PMNs.

The migration of directly CAP-treated PMNs was analyzed in 30 min periods.

In the 4 kHz group, the period from Minute 1 to 30 did not show any difference in the Track Length of the control group and 2 min of CAP treatment, whereas 5 min of CAP treatment lowered the Track Length significantly (*p* = 0.0296) (Figure 1B(I)). Track Displacement X decreased in comparison to the control group after both CAP treatment durations (*p*_2min_ = 0.0144; *p*_5min_ = 0.0002) (data not shown). The 8 kHz CAP treatment also decreased the Track Length after 5 min in comparison with the control group (*p* = 0.0161) (Figure 1B(II)), but Track Displacement X did not show any differences.

In the second observation period from Minute 31 to 60, the 4 kHz CAP treatment again reduced the Track Length after 5 min compared to the control group (*p* = 0.0021) (Figure 1B(III)) but had no effect on Track Displacement X. In comparison with the control group, 8 kHz CAP treatment prolonged the Track Length after 2 min (*p* = 0.0267) (Figure 1B(IV)) and Track Displacement X after both treatment durations (*p*_2min_ = 0.0012; *p*_5min_ = 0.0172).

Between Minutes 61 and 90, the Track Length of the CAP-group treated with 4 kHz for 2 min was lengthened compared to the control group (*p* = 0.0011), whereas 5 min of CAP treatment again reduced the Track Length (*p* = 0.0012) (Figure 1B(V)). The Track Displacement X was only decreased after 5 min of CAP treatment (*p* = 0.0189). The extension of the Track Length after 2 min of 8 kHz CAP treatment was re-monitored in comparison with the control group (*p* = 0.0028) (Figure 1B(VI)), but Track Displacement X was not altered.

Taken together, PMN migration differed according to treatment duration and mode. Whereas 2 min of CAP treatment in both modes prolonged the Track Length, just 5 min of 4 kHz treatment decreased the Track Length in the long term. The effect on Track Displacement X was by trend decreasing after 4 kHz treatment and increasing after 8 kHz treatment.

### 3.2. Altered T_max_ROS without Affecting NETosis after CAP Treatment

Live-cell imaging during microscopy was used to assess T_max_ROS and ET_50_NETosis. T_max_ROS is the timepoint where most of the cells are detected as ROS containing. ET_50_NETosis is defined as the timepoint when 50% of all PMNs that underwent NETosis reached the final state of extracellular DNA staining with DAPI. Both parameters could be influenced by CAP treatment and thus take place earlier or later. To compare the treatment groups, all values were standardized to their respective control group median.

The T_max_ROS of the PMNs was reduced after 5 min of direct 4 kHz CAP treatment of the PMNs compared to the control group (*p* = 0.0187) (Figure 2A(I)), whereas 8 kHz CAP treatment did not influence T_max_ROS (Figure 2A(II)). ET_50_NETosis was not affected by any type of CAP treatment (Figure 2B(I,II)). The effects of 4 kHz treatment on T_max_ROS were also observed in experiments using 2 min or 5 min of PTS (*p*_2min_ = 0.0029; *p*_5min_ = 0.0002) (data not shown).

These results imply that the time course of granulocyte functions can only be partly manipulated with CAP and is influenced by the CAP mode and treatment duration.

### 3.3. Increase in ROS and RNS Concentrations in PTS

To examine which components of PTS are modified, the contained ROS and RNS were measured without CAP treatment, after 5 min of 4 kHz treatment and 5 min of 8 kHz treatment of the medium. ROS were scaled by the median fluorescence intensity (MFI) of Rhodamine123 and the concentration of H_2_O_2_. RNS were determined by measuring NO_2_^−^ and NO_3_^−^ concentrations.

The measurements showed induction of the MFI of Rhodamine123 (*p* = 0.0010) (Figure 3I) and an increased concentration of H_2_O_2_ (*p* = 0.0016) (Figure 3II) after the oxygen treatment mode. The nitrogen treatment mode showed rather stimulating effects on NO_2_^−^ (*p* = 0.0002) (Figure 3III) and NO_3_^−^ (*p* = 0.0463) (Figure 3IV) as well as on H_2_O_2_ (*p* = 0.0288) (Figure 3II).

These results imply that the two treatment modes clearly differ in their amount of ROS and RNS production.

### 3.4. Changed Activity of Respiratory Burst after CAP Treatment

The respiratory burst of PMNs, which indicates the intensity of ROS production, was caused either by fMLP and TNFα or by PMA and was quantified by flow cytometry. Whereas fMLP and TNFα created a moderate receptor-mediated stimulus, PMA activated the protein kinase c directly and was used as a positive control. The aim was to identify differences in ROS production between treatment groups and stimulations to show a potential impact by CAP treatment.

Two hours after CAP treatment, stimulation with fMLP and TNFα had significantly increased the MFI of Rhodamine123 in the 8 kHz CAP group treated for 5 min compared to the MFI of the control group (*p* = 0.0107) (Figure 4I). Stimulated with fMLP and TNFα and compared to the MFI of the control group, the MFI of Rhodamine123 was significantly decreased in both 4 kHz groups 6 h after CAP treatment (*p*_2min_ = 0.0070; *p*_5min_ = 0.0024), but 8 kHz treatment had no effect (Figure 4II). After PMA stimulation, the groups treated with 4 kHz showed no change. The group treated with 8 kHz for 5 min showed a significant increase in the MFI of Rhodamine123 2 h after CAP treatment in contrast to the control group (*p* = 0.0008) (Figure 4III). Both groups treated with 8 kHz showed the observed effect 6 h after CAP treatment compared to the control group (*p*_2min_ = 0.0177; *p*_5min_ = 0.0002) (Figure 4IV). Representative histograms of the flow cytometry data can be found in Appendix A.

These results suggest the presence of mode- and stimulation-dependent alterations in respiratory burst activation after CAP treatment.

### 3.5. PMN Membrane Permeability for PI after CAP Treatment

Membrane permeability was quantified 2 and 6 h after CAP treatment by the flow cytometric measurement of the percentage of PI-positive PMNs. This was done to investigate the influence of CAP on cell membranes.

In general, we observed a certain donor dependency in PI-positive cells. 8 kHz CAP treatment for 5 min significantly increased PI-positive cells 2 h after treatment compared to the control group (*p* = 0.0024) (Figure 5I). Six hours after CAP treatment, the percentage of PI-positive cells did not show any significant changes (Figure 5II).

This finding leads to the assumption that membrane permeabilization of PMNs is not stimulated by CAP in the long term.

### 3.6. Promoted Integrin and Selectin Expression on PMN Surfaces

The expression of the cell-surface antigens CD11b, CD62L, and CD66b on PMNs was quantified by flow cytometry 2 h and 6 h after CAP treatment of the cells to detect potential cell activation.

CD11b expression on PMNs measured 2 h (Figure 6A(I)) and 6 h (Figure 6A(II)) after CAP treatment was only significantly increased 2 h after 4 kHz treatment for 2 min compared to the control group (*p* = 0.0451). In contrast, CD11b expression was significantly increased 2 h as well as 6 h after 4 kHz treatment for 5 min (*p*_2h_ <0.0001; *p*_6h_ = 0.0177) and 8 kHz treatment for 5 min (*p*_2h_ = 0.0014; *p*_6h_ = 0.0177).

CD62L expression showed exactly the opposite reaction because it was significantly decreased after 4 kHz CAP treatment for 5 min (*p*_2h_ = 0.0451; *p*_6h_ = 0.0006) and after 8 kHz CAP treatment for 5 min compared to the control group (*p*_2h_ = 0.0070; *p*_6h_ = 0.0456). Treatment with 4 kHz for 2 min also tended to decrease values, but these trends were not significant (Figure 6B(I,II)).

In comparison with the control group, CD66b expression was enhanced in both measurements after 4 kHz CAP treatment for 5 min (*p*_2h_ <0.0001; *p*_6h_ = 0.0020), and after 8 kHz CAP treatment for 5 min (*p*_2h_ = 0.0001; *p*_6h_ = 0.0107). Further, the expression was heightened 2 h after 4 kHz CAP treatment for 2 min (*p* = 0.0451) (Figure 6C(I,II)).

Compared to the integrin and selectin expression of the untreated control groups, CAP treatment increased CD11b and CD66b and decreased CD62L. This constellation suggests the activation of PMNs.

## 4. Discussion

The study analyzed the functionality of human polymorphonuclear cells (PMNs) modified through either PTS or direct CAP treatment with frequencies of 4 or 8 kHz and 2 or 5 min treatment durations. We aimed to analyze how these different CAP treatment modalities may impact PMN immunoreactivity and how these changes may influence wound healing. It is important to note that, in this preclinical study, no direct investigations were made in the context of wound healing, and all connections in this direction are speculative. However, our in vitro analyses on primary isolated PMNs from human donors impressively showed that by changing the CAP parameters (frequency or time) and the treatment modality (directly or indirectly), activating effects on granulocytes could be observed. These effects, depending on the clinical context, can be both beneficial and inhibitory, for example, in acute or chronic wounds or in pyoderma gangraenosum. Therefore, no conclusive recommendation can be made from this study as to which treatment modalities promote wound healing per se.

Wound healing normally consists of four stages: hemostasis, inflammation, proliferation, and remodeling. These stages can be influenced by CAP and thus make a final contribution to improved wound healing. Several studies have suggested CAP technology to be a useful tool for promoting hemostasis [28,29]. Furthermore, improved proliferation and remodeling after CAP treatment were observed [13]. Our study, therefore, concentrated on inflammation and investigated the immunoreactivity of PMNs in vitro. PMNs—the most abundant inflammatory cells—are activated by damage-associated molecular patterns and migrate towards the site of injury [30]. PMNs can eradicate pathogens through ROS release during the oxidative burst and by NET formation. These functions can be regulated by CAP treatment [31,32]. NETs consist of chromatin, histonic and non-histonic proteins, and microbicidal agents. During NETosis, which is regarded as an extreme defensive mechanism, NETs play also a critical role in wound healing. Low NET concentrations support keratinocyte proliferation and consequently wound healing, whereas high concentrations induce an opposite effect [33]. Physiologically, PMNs undergo apoptosis and phagocytosis after conducting their functions, which marks the end of the inflammatory phase. The prolonged presence of granulocytes with the excessive production of ROS and NETs is associated with chronic lesions [16]. Hence, we assumed that increased activation and migration of PMNs may be beneficial in CAP-induced wound healing, but that excessive ROS production and NET formation or late apoptosis may be counterproductive.

It has to be mentioned that the composition of CAP depends on the plasma source used. In this study, we used a surface micro-discharge (SMD) plasma source operating at low voltages of 3.5 kV. The energy output of plasma sources is directly associated with lethal effects on cells [34]. Furthermore, different treatment durations have different CAP effects on different cell types [8,34,35]. In this study, a treatment time of 2 min was defined as a short treatment duration and 5 min as a long treatment duration.

CAP effects on different cell types have already been investigated, including alterations of migration, intracellular ROS, plasma membrane, and cell viability [34,36,37,38]. We based our study on these commonly known cellular plasma effects, as there are hardly any studies on this with PMNs. However, PMNs play an important role during inflammatory processes and therefore require more detailed analysis.

PMN involvement in inflammatory processes starts with their extravasation, which follows a defined recruitment cascade including rolling, adhesion, crawling, and transmigration. Rolling is mainly selectin-dependent, whereas adhesion, crawling, and transmigration depend on integrin interaction [39,40]. However, the selectin CD62L is eliminated from the plasma membrane of PMNs through ectodomain shedding, which, in turn, is an indicator for granulocyte activation according to the upregulation of integrin CD11b expression. CD66b, a PMN immunoreceptor, also indicates activation by upregulation [41]. In our study, we investigated the expression of these surface markers. Compared to the expression of integrin and selectin in the untreated control groups, particularly 5 min of CAP treatment with either 4 or 8 kHz showed a distinct upregulation of CD11b and CD66b and downregulation of CD62L 2 and 6 h after treatment. This constellation suggests the CAP-mediated activation of PMNs. The increased activation also leads to the expectation that the subsequent processes of migration, ROS production, and NETose will run faster, but not necessarily more strongly. PMN trafficking towards inflammation starts with the expression of the selectin CD62L [42]. Therefore, significantly reduced expression of CD62L induced by CAP treatment and subsequent upregulation of CD11b and CD66b may be assumed to improve PMN migration to the site of infection.

The further experimental setup simulated extravascular PMN migration towards the bacteria-derived chemo-attractant fMLP after extravasation. fMLP is known as a potent chemo-attractant, which leads PMNs to the site of infection [39]. The exchange of fMLP with PTS significantly lowered Track Displacement X and the Track Length of PMNs. Thus, PTS is unlikely to have any chemo-attractive effect on PMNs.

The migration data of direct CAP-treated PMNs showed different PMN migration after different treatment durations and modes. Even the 4 kHz and 8 kHz control groups showed alterations; however, PMNs are circulating cells of the immune system, so interindividual differences in their behavior are possible [43,44]. Two minutes of CAP treatment in both modes prolonged the Track Length, whereas only 5 min of 4 kHz treatment lowered the Track Length and thus reduced migration in the long term. These findings are in line with the generally described effects of CAP treatment on cells, such as proliferation, migration, or cell death, which are stimulated after short treatment but inhibitory after longer treatment durations [34]. Concerning migration, this effect had already been observed in keratinocytes treated with PTS that showed elevated migration after 15–30 s of CAP treatment and degraded after longer treatments of 45–90 s [9]. Similar modulations leading to enhanced migration were noticed in fibroblasts after 30 s of CAP treatment [13]. In addition, the shortened migration after 5 min of 4 kHz CAP treatment fits the assumption that the subsequent processes run faster after increased activation since we measured increased activation after 5 min of CAP treatment. However, in our study, 8 kHz treatment did not decrease the Track Length. The effect on Track Displacement X was by trend decreased after 4 kHz treatment and increased after 8 kHz treatment. These observations led to the assumption that, next to a particular treatment duration depending on modulation of the Track Length, the treatment mode is also a vital part of PMN migration that especially modifies Track Displacement X. Because Track Displacement X is an indicator for PMN movement towards a chemo-attractant, CAP can be assumed to be able to modify this behavior.

The different effects of the two treatment modes can be explained by reference to the measurements of reactive species in PTS, which were also investigated in this study. We have proved with our PTS measurements that the two treatment modes 4 kHz and 8 kHz significantly differ in their amount of ROS and RNS production. As the 4 kHz mode particularly produces ROS and the 8 kHz mode RNS, it can be assumed that the migration of PMNs is accelerated by ROS and slowed down by RNS. Compatible with this hypothesis, a previous study showed that H_2_O_2_ stimulated leucocyte recruitment to injured tissue [45].

Adequate ROS concentrations in wound tissue, however, are important for cellular homeostasis [46]. Furthermore, a prolonged presence of PMNs with subsequent production of ROS leads to the destruction of cells, thus contributing to the formation of chronic lesions by causing further production of inflammatory mediators [16]. On the other hand, nitric oxide leads to stimulation and phagocytosis of bacteria and necrotic detritus by granulocytes [47]. Our respiratory burst measurements revealed mode- and stimulation-dependent alterations in respiratory burst activation after CAP treatment. We measured decreasing activity after 4 kHz treatment combined with fMLP and TNFα stimulation and increasing activity after 8 kHz treatment combined with fMLP and TNFα or PMA stimulation. Based on these findings, it may be hypothesized that reduced PMN ROS production after CAP treatment is beneficial in wound healing in preventing excessive ROS concentrations in wound tissue because CAP alone already contains ROS.

The assessment so far correlates in part with the results concerning T_max_ROS, which was significantly lowered after 5 min of 4 kHz CAP treatment but not affected after 2 min of 8 kHz treatment. Again, in the case of the earlier T_max_ROS after 5 min of 4 kHz CAP treatment, this is compatible with the hypothesis that increased activation levels lead to faster subsequent processes. As after increased activation, the migration was shortened, it is conclusive that the T_max_ROS also takes place earlier. Interestingly and against expectations, ET_50_NETosis could not be manipulated at all. In a previous study, however, PMNs incubated with exogenously added peroxynitrite showed induced NET formation in a concentration-dependent manner [48]. Our results, therefore, suggest that the time course of PMN functions may only be partly manipulated with CAP and might depend on the plasma source. The plasma parameters and our treatment approach will need to be further modified to observe any effects.

Furthermore, propidium iodide (PI)-positive PMNs were significantly increased 2 h after 5 min of 8 kHz treatment, but this effect was no longer existent 6 h after CAP treatment. This finding leads to the assumption that CAP treatment temporarily induces PMN membrane permeability for PI; therefore, 2 h after CAP treatment, the percentage of PI-positive cells does not coincide with the actual percentage of dead cells. This assumption stands in line with an already existing study indicating a CAP-induced uptake mechanism of PI and anti-cancer agent in cancer cells without affecting viability [49]. Additionally, a CAP-mediated increase in intracellular ROS may induce membrane permeability [50]. This assumption corresponds with our observance of an increased respiratory burst after 5 min of 8 kHz CAP treatment and fMLP and TNFα stimulation that was only noticeable 2 h after treatment. Therefore, we concluded that CAP treatment does not have any apoptosis-inducing effect on PMNs in the long term. Nevertheless, we quantified PI-positive cells, which indicates membrane permeabilization in general but no differentiation between the different grades of apoptosis [51].

Taking all the discussed results together, our study indicates that the immunoreactivity of PMNs can be influenced by CAP treatment, which we investigated by measuring a variety of parameters and summarized in Table 1. For an improved overview, we summarized the observed CAP-modified processes in a timeline. Firstly, PMNs are activated. As a side effect, the membrane permeability is temporarily increased. Secondly, depending on the intensity of activation, which is determined by the treatment duration, the cells show either increased migration in the case of lower activation or decreased migration in the case of high activation. We assume that high activation levels lead to a faster final activation of ROS production, which implements the stop of migration. Thirdly, the intensity of ROS production is dependent on the treatment mode as well as the cell stimulation and can be increased or decreased. In our study, we observed no CAP influence on the final NET formation and viability. Further in vivo experiments are necessary to fully explore the influence of CAP on the immunoreactivity of PMNs and its impact on wound healing during the inflammatory phase.

## 5. Conclusions

Although the use of CAP in medicine and especially in the treatment of chronic and infected wounds is rather common, its possible repercussions and induced cellular mechanisms need to be thoroughly assessed. Our study contributes to the research on CAP interaction with PMNs. In summary, our results show that migration is generally improved by 2 min of CAP treatment. Concerning other parameters, the effects of CAP treatment need to be distinguished according to the used treatment modes. A 4 kHz treatment generally determines PMN activation, lowers respiratory burst activity after fMLP and TNFα stimulation, and reduces T_max_ROS after 5 min of treatment. Five minutes of 8 kHz treatment increases respiratory burst after PMA stimulation in addition to PMN activation (Table 1). Although CAP has a variety of effects on PMNs, it is not possible to define one treatment mode and duration as the most beneficial in the context of wound healing. Nevertheless, regarding improved PMN functions, a long treatment duration seems to amplify activation of the cells, whereas short treatment increases their migration. Cellular ROS production, which is critical in high amounts in chronic wounds, can be diminished with the 4 kHz oxygen treatment mode. On the supposition that this constellation of modified PMN functions might improve wound healing, short treatment with the oxygen mode combines most of the beneficial effects of CAP treatment. With this knowledge, further investigations should examine the in vivo effects of CAP treatment on PMNs after wound treatment to gain more knowledge about further particulars concerning wound healing.

## Figures and Tables

**Figure 1 biomolecules-11-00902-f001:**
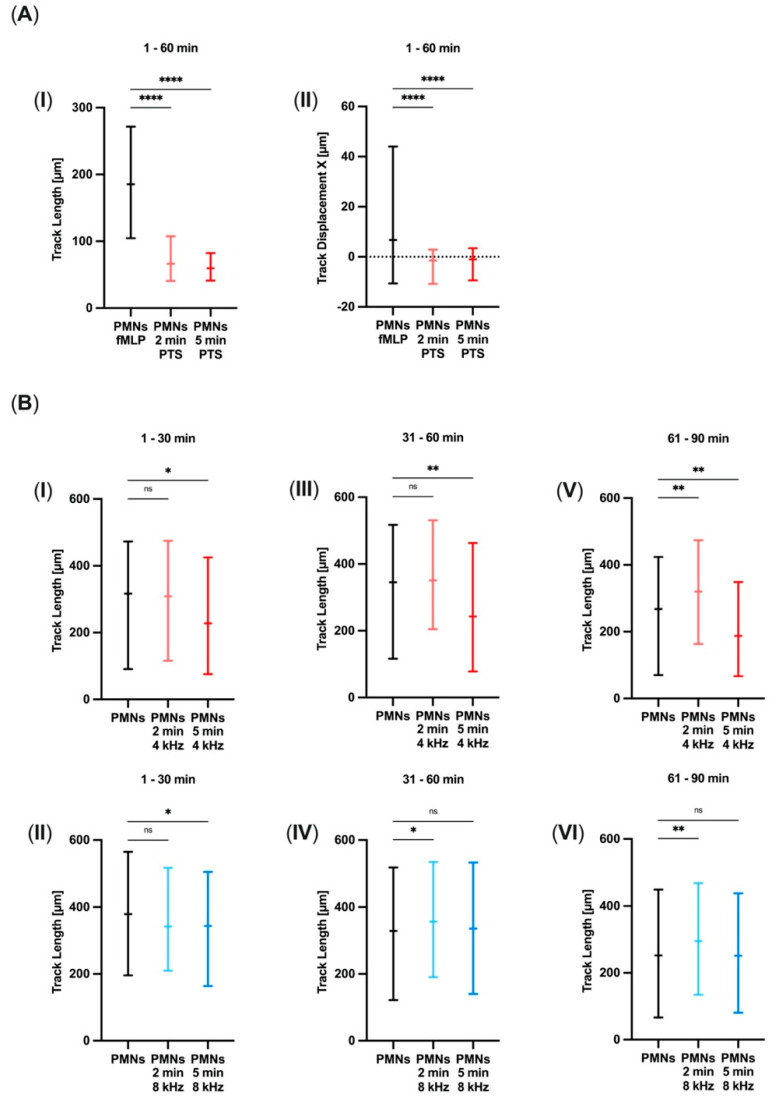
Modified migration of CAP-treated PMNs without any chemo-attractive PTS effect. (**A**, **I**, **II**) Track Length and Track Displacement X [µm] of PMNs towards either the chemo-attractant fMLP or 2 min or 5 min plasma-treated solution (PTS) during Observation Minutes 1–60 (*n* = 5; mean number of tracks = 882). (**B**, **I**–**VI**) Track Length [µm] of PMNs after CAP treatment with 4 kHz (*n* = 6; mean number of tracks = 387) or 8 kHz (*n* = 5; mean number of tracks = 814) for 2 min and 5 min in 30-minute observation periods from Minute 1 to 90. Migration was quantified by semiautomated cell tracking during live cell imaging. Statistical analysis: Kruskal–Wallis test. * *p* < 0.05, ** *p* < 0.01, **** *p* < 0.0001, ns: not significant.

**Figure 2 biomolecules-11-00902-f002:**
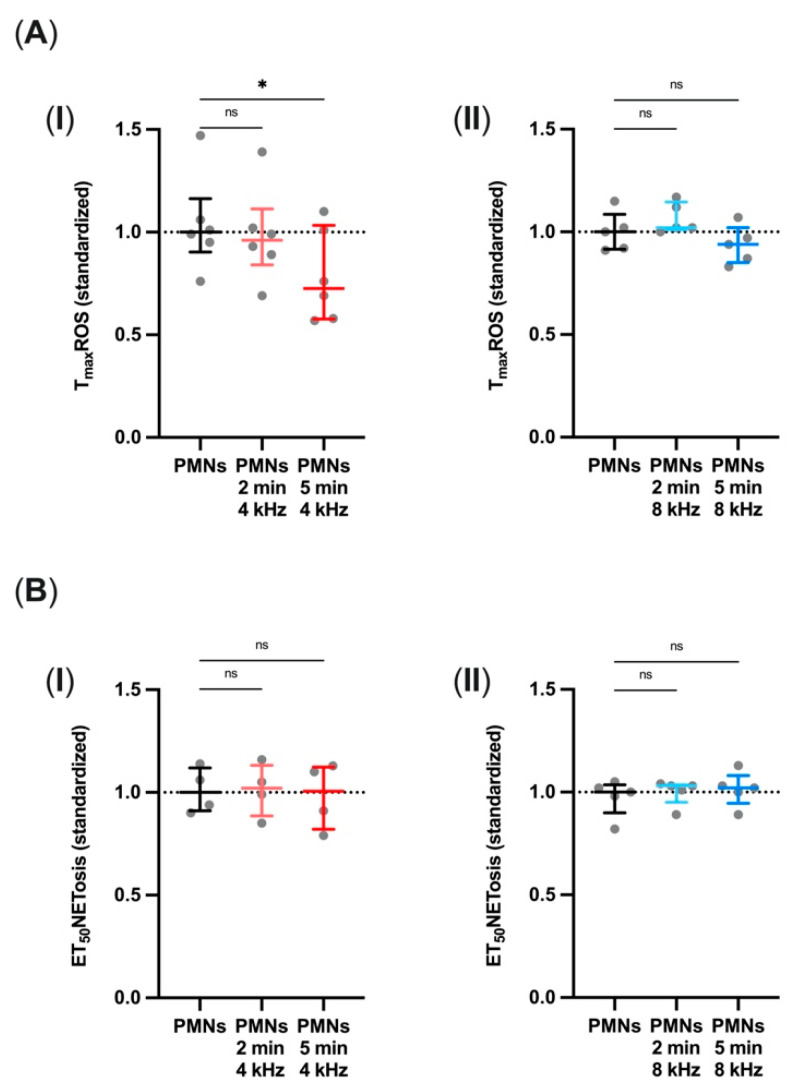
Altered T_max_ROS without affecting NETosis after CAP treatment. (**A**, **I**, **II**) Standardized T_max_ROS after 4 kHz (*n* = 6) or 8 kHz (*n* = 5) CAP treatment for 2 min and 5 min. (**B**, **I**, **II**) Standardized ET_50_NETosis after 4 kHz (*n* = 4) or 8 kHz (*n* = 5) CAP treatment for 2 min and 5 min. Both parameters were evaluated by processing the total fluorescent areas per time point during live cell imaging. Statistical analysis: Friedman test. * *p* < 0.05, ns: not significant.

**Figure 3 biomolecules-11-00902-f003:**
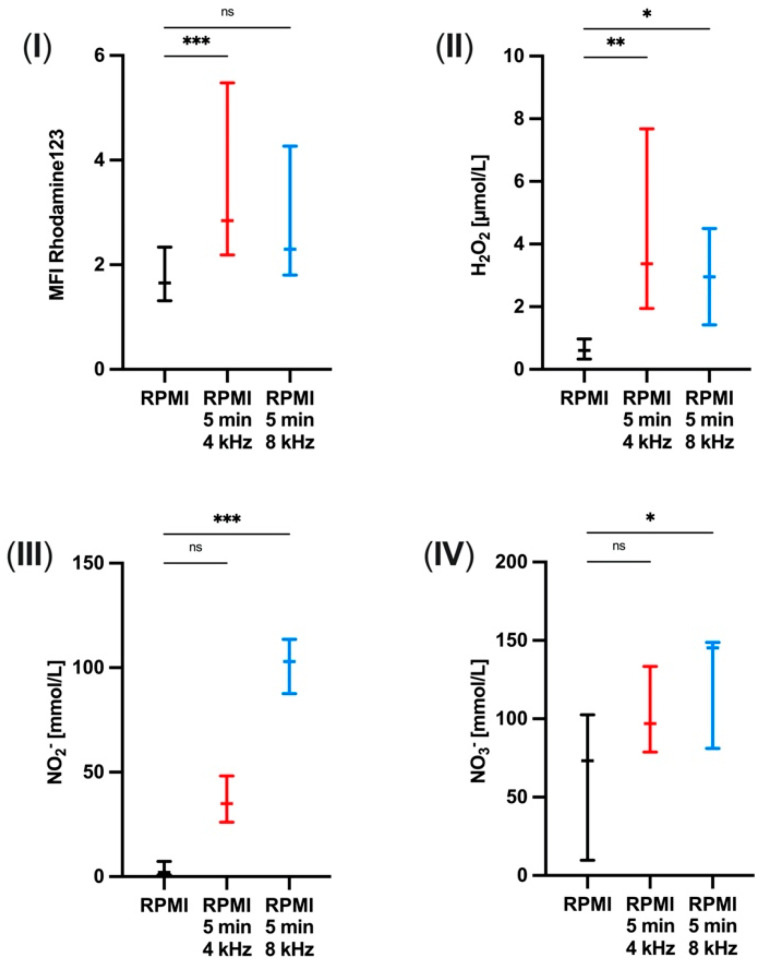
Increase in ROS and RNS concentrations in the plasma-treated solution. (**I**) Median fluorescence intensity (MFI) of Rhodamine123, (**II**) the concentration [µmol/L] of H_2_O_2_, (**III**) the concentration [mmol/L] of NO_2_^−^, and (**IV**) the concentration [mmol/L] of NO_3_^−^ in RPMI 1640 dosed with 10% fetal bovine serum after 5 min of 4 kHz and 8 kHz CAP treatment (*n* = 4, each with 3 technical replicates) detected by fluorescence measurements. Statistical analysis: Kruskal–Wallis test. * *p* < 0.05, ** *p* < 0.01, *** *p* < 0.001, ns: not significant.

**Figure 4 biomolecules-11-00902-f004:**
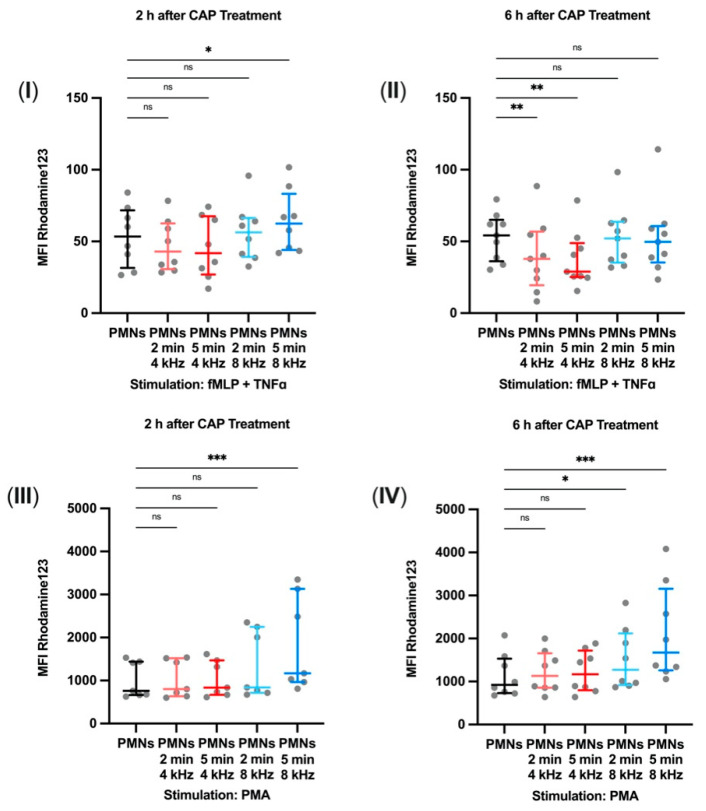
Changed activity of respiratory burst after CAP treatment. PMN respiratory burst activity after CAP treatment with 4 kHz or 8 kHz for 2 min and 5 min, measured by flow cytometry as median fluorescence intensity (MFI) of Rhodamine123. fMLP and TNFα stimulation (**I**) 2 h (*n* = 8) and (**II**) 6 h (*n* = 9) after CAP treatment. PMA stimulation (**III**) 2 h (*n* = 7) and (**IV**) 6 h (*n* = 8) after CAP treatment. Statistical analysis: Friedman test. * *p* < 0.05, ** *p* < 0.01, *** *p* < 0.001, ns: not significant.

**Figure 5 biomolecules-11-00902-f005:**
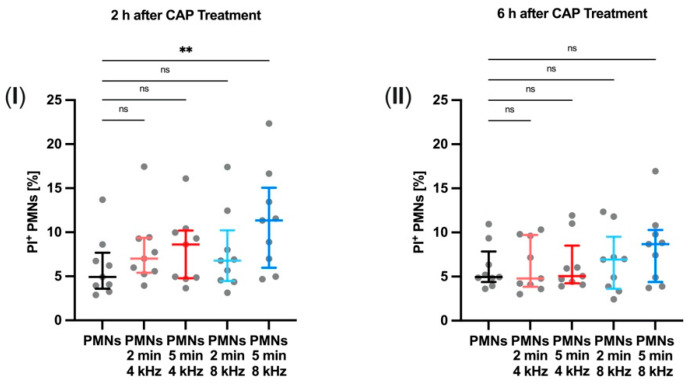
PMN membrane permeability for propidium iodide (PI) after CAP treatment. The percentage of PI-positive PMNs evaluted by flow cytometry (**I**) 2 h and (**II**) 6 h after 4 or 8 kHz CAP treatment for 2 and 5 min (*n* = 9). Statistical analysis: Friedman test. ** *p* < 0.01, ns: not significant.

**Figure 6 biomolecules-11-00902-f006:**
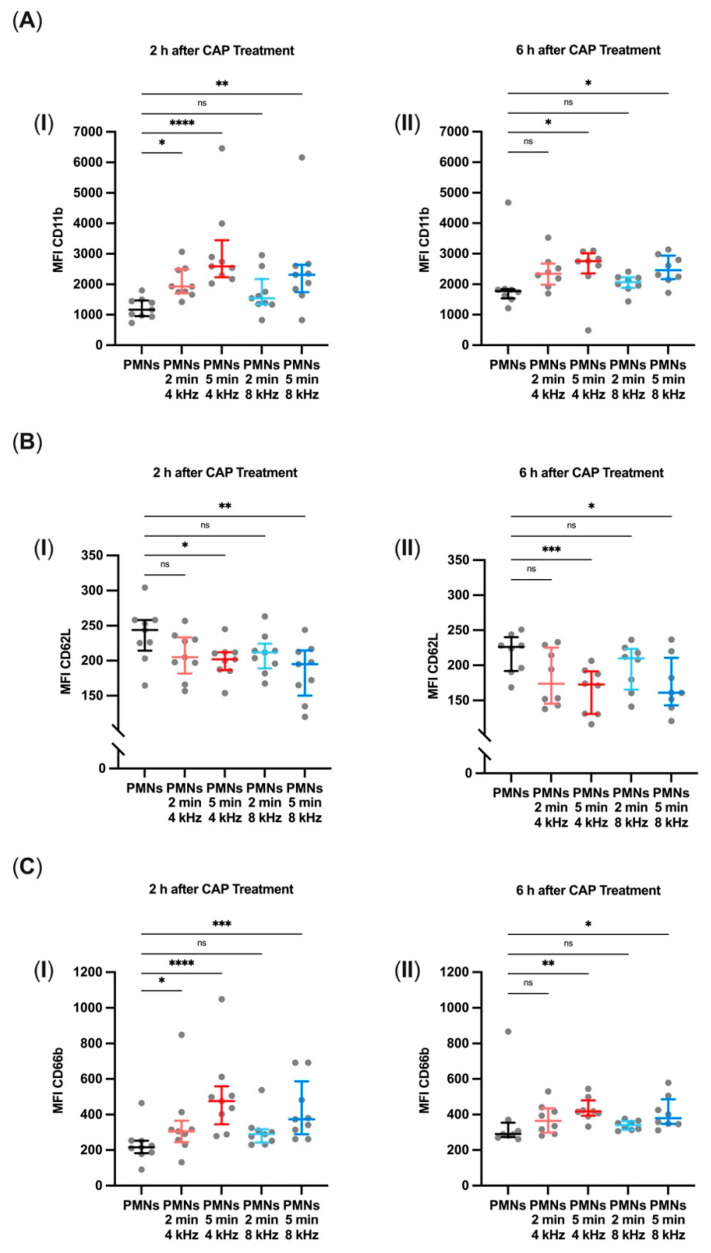
Promoted integrin and selectin expression on PMN surfaces. Flow cytometric measurement of the median fluorescence intensity (MFI) of (**A**, **I**, **II**) CD11b, (**B**, **I**, **II**) CD62L, and (**C**, **I**, **II**) CD66b 2 h (*n* = 9) and 6 h (*n* = 8) after CAP treatment with 4 kHz or 8 kHz for 2 min and 5 min. Statistical analysis: Friedman test. * *p* < 0.05, ** *p* < 0.01, *** *p* < 0.001, **** *p* < 0.0001, ns: not significant.

**Table 1 biomolecules-11-00902-t001:** Overview of CAP effects on PMNs divided into 4 and 8 kHz treatment modes and 2 and 5 min treatment durations.

CAP Treatment Mode	4 kHz Oxygen Mode		8 kHz Nitrogen Mode	
Duration	2 min	5 min	2 min	5 min
Migration	↑↑	↓↓	↑↑	↓
Induction period of ROS production	↔	↓↓	↔	↔
Intensity of ROS production after fMLP and TNFα stimulation	↓↓	↓↓	↔	↑
Intensity of ROS production after PMA stimulation	↔	↔	↑↑	↑↑
Duration until NETosis onset	↔	↔	↔	↔
Membrane permeability (2 h after CAP treatment)	↔	↔	↔	↑↑
Membrane permeability (6 h after CAP treatment)	↔	↔	↔	↔
Activation (indicated by integrin and selectin expression)	↑	↑↑	↔	↑↑

Note: ↑/↓ indicates moderate change, ↑↑/↓↓ indicates significant change, ↔ indicates no change compared to untreated PMNs (*p* < 0.05).

## Data Availability

The data presented in this study are available on request from the corresponding author.

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
