# Peer review of "Cold Atmospheric Plasma Promotes the Immunoreactivity of Granulocytes In Vitro"

_biomolecules, 2021, doi:10.3390/biom11060902_

Round 1

Reviewer 1 Report

Please see attached review.

Author Response

Please see attached PDF file.

Reviewer 2 Report

Manuscript reference number: Biomolecules-1211653

Summary:

The manuscript “Cold Atmospheric Plasma Induces Granulocyte Function in Context of Wound Treatment” is an original article that evaluates the effects of different cold atmospheric plasma (CAP) application strategies on the cellular function of human polymorphonuclear neutrophils (PMN) using cell culture experiments. They used a jet device to apply direct CAP treatment on cell cultures using different plasma modes (4 kHz and 8kHz) and exposure lengths (2 min and 5 min). Biological (cell migration and vascular adherence/rolling capacity, survival, membrane permeability) and molecular [reactive oxygen species (ROS), reactive nitrogen species (RNS)] effects were assessed at different times (2h, 6h) after CAP treatment.

The authors observed that CAP can stimulate the function of PMNs by facilitating their ability to migrate, adhere, or rolling, through ROS production and a transient increase in membrane permeability, without survival reduction by apoptosis/NETosis. These effects

The manuscript is very well written; clear, precise, and easy to understand. The methodology is acceptable and correct.

However, in my opinion, some questions should be addressed by authors before a final decision is reached.

1: Authors talk about CAP effect on wound biology, but this study is only focused on assessing PMN function (inflammation) and there is no experiment in the methodology that evaluated the ability of CAP to improve other specific steps of wound healing: blood clotting (hemostasis), tissue growth (cell proliferation), and tissue remodeling (maturation and cell differentiation). We could say that the design and results are more general and no so specifically to wound-healing, so they should clarify this point throughout the article (and title).

2: This study could be considered as a dose-effect study to find to judge the efficacy and safety of different doses of CAP at the cellular level. However, the authors do not make a clear recommendation about which combination of CAP modality and exposure time would be most useful to use in further studies, in light of their results.

3: How they explain no PMN activation after 8 kHz CAP exposure during 2 min? This is especially relevant when on the contrary this effect was observed using lower doses and time of exposure (4 kHz, 2 min). Please, clarify.

4: The content of a footnote on table 1 seems controversial. The original footnote displays: ↑/↓ indicates temporary change, ↑↑/↓↓ indicates significant change over time. What do 'temporary' and 'significant' mean? Does 'temporary mean' for a short period? - and then 'significant' means for a 'large period'? Or does 'significant' mean 'statistically significant? - and then 'temporary' means 'not statistically significant?

5: How do you interpret a significant positive dose-effect of PMN activation but a negative in cell migration after 5 min vs 2 min of CAP exposure?

Reviewer 3 Report

The manuscript by Kupke et al. entitled ‘Cold Atmospheric Plasma Induces Granulocyte Function in Context of Wound Treatment’ is a well-written article in which the author performed most key experiments the topic. In this article, the author used a cold-plasma technique that induces RONS, and here it modulates the granulocytes function (neutrophils only) in the wound treatment. However, many key experiments are missing, and the author lacks to explain many things, which makes this article unsuitable for publication in its present form. I encouraged the author, refine the article once again. Therefore, I rejected the article for publication in this journal. However, the following points need to be included for further submission in another journal.

  1. The main title said ‘Granulocyte function’; however, the author described only neutrophils in the whole article. Therefore, this must be changed to ‘Neutrophils function’ in the title.
  2. Many key articles were not cited in the whole article. A few of them are as follows.

  • Sander Bekeschus, Christine C Winterbourn, Julia Kolata, Kai Masur, Sybille Hasse, Barbara M Bröker, Heather A Parke. Neutrophil extracellular trap formation is elicited in response to cold physical plasma. J Leukoc Biol. 2016 Oct;100(4):791-799. DOI: 10.1189/jlb.3A0415-165RR.

  • Boeckmann et al. 2020. Cold Atmospheric Pressure Plasma in Wound Healing and Cancer Treatment. Appl. Sci. 2020,10, 6898; doi:10.3390/app10196898.

  1. The author's key factors in wound healing are platelets and Vitamin K, which were totally neglected and didn’t explain. The author strongly suggests performing or discuss some experiments related to the modulation of these factors on cold plasma-induced wound healing.
  2. All the experiments were performed without the inclusion of positive control. However, positive control must be included in some of the crucial experiments e., Figure 3.
  3. The X-axis legends of Figure 3 are not consistent with the rest of the figures. It must be RPMI <Time> <Frequency> in all figures.
  4. The author strongly advises including flow cytometry data for figure 4 (in the histogram). Although the difference is statistically non-significant, the inclusion of the flow cytometry figure will better understand the outcome.
  5. Figure 6 need to be changed. The parameter at Y-axis is very high, which makes the difference unable to differentiate between each group. It is recommended to keep MFI values within the minimum value range so that the differences can be visible.
  6. How the neutrophils segregate from the granulocytes? Since CD66b also expressed by eosinophil, how does the author confirm that CD66b signaling is only neutrophils?
  7. The Optical Emission Spectrum (OES) of the cold plasma should be included. This is because the RONS generated by the cold plasma solely depends on the gas type, moisture, and plume to sample distance. Therefore, the amount of RONS generated by the cold plasma directly relates to the RONS generated in the medium (RPMI). Therefore, the OES of the device must be presented every time or provide the reference if the same device was used before in any publication.

Author Response

Please see attached PDF file.

Round 2

Reviewer 1 Report

Thank you for addressing the comments but there are still several issues. Please see attached.

Reviewer 2 Report

All request questions have been addressed.

Reviewer 3 Report

The author thoroughly revised the manuscript according to the raised queries. The manuscript is now looked good to be accepted. 

Round 3

Reviewer 1 Report

Thank you for your responses. While I still have some hesitation, I believe the authors have made an honest attempt to address my concerns. I look forward to further work from this team.